**Invited perspective - Redefining Disaster Risk: The Convergence of Natural Hazards and Health Crises**

Nivedita Sairam[1] and Marleen C. de Ruiter[2]

[1] Section Hydrology, GFZ German Research Centre for Geosciences, Potsdam, Germany

[2] Institute for Environmental Studies, Vrije Universiteit Amsterdam, De Boelelaan 1111, 1081HV, Amsterdam, the
Netherlands

*Correspondence to*: Marleen de Ruiter (m.c.de.ruiter@vu.nl)

**Abstract.** Recently, the disaster risk field has made substantial steps forward to develop increasingly comprehensive risk assessments, accounting for the incidence of multiple hazards, trickle-down effects of cascading disasters and/or impacts, and spatiotemporal dynamics.
While the COVID-19 outbreak increased general awareness of the challenges that arise when disasters from natural hazards and diseases collide, we still lack a comprehensive understanding of the role of disease outbreaks in disaster risk assessments and management, and that of health impacts of disasters. In specific, the occurrence probabilities and the impacts of disease outbreaks following natural hazards are not well-understood and are commonly excluded from multi-hazard risk assessments and management.
Therefore, in this perspective paper, we develop a research agenda that focusses on 1. learning lessons from interdisciplinary communities such as compound risks and the socio-hydrology community for modelling the occurrence probabilities and temporal element (lag times) of disasters and health/disease-outbreaks, 2. the inclusion of health-related risk metrics within conventional risk assessment frameworks, 3. improving data availability and modelling approaches to quantify the role of stressors and interventions on health impacts of disasters. Collectively, this agenda is intended to advance our understanding
of disaster risk considering potential health crises. The developed research agenda is not only crucial for scientists aiming to improve risk modelling capabilities, but also for decision makers and practitioners to anticipate and respond to the increasing complexity of disaster risk.

## 1 Introduction

On August 14, 2021, a 7.2 Mw earthquake struck Haiti's southern peninsula, followed by smaller aftershocks, including a 5.8
Mw earthquake. The earthquake caused widespread landslides and rockfalls, damaging roads and isolating communities (Cabas et al., 2023). It resulted in over 12,000 injuries, more than 300 missing, and at least 2,248 deaths, with 137,000 homes destroyed or severely damaged. Key infrastructure, including schools, churches, bridges, and roads, was also impacted, disrupting access to education, water, sanitation, and healthcare services (CDEMA, 2021; GoH, 2021). As many remained outdoors due to damaged homes and aftershock fears, tropical storm Grace struck on August 16-17, causing heavy rainfall,
winds, flash flooding, and landslides, which halted rescue efforts for hours (Cavallo et al., 2021; Reinhart & Berg, 2022). The storm's impact, compounded by the earthquake, made it difficult to distinguish the sources of casualties (Reinhart & Berg, 2022). Initial aid was delayed due to the remote and inaccessible regions affected (Cabas et al., 2023; Daniels, 2021). The destruction of WASH (Water, Sanitation and Hygiene) infrastructure and healthcare facilities increased the risk of waterborne diseases, contributing to a cholera outbreak a year later. By November 2022, over 230 people had died from cholera, with
12,500 suspected cases (IFRC, 2022). Additionally, many people, forced to sleep outside or in inadequate shelters, were vulnerable to storm-related hazards and aftershocks (Daniels, 2021; OCHA, 2021).

In contrast to these acute disasters in Mozambique, during 2017-2018, Kenya and Ethiopia were exposed to slow onset, chronic disasters caused by back-to-back hydrological extremes. A severe drought (Funk et al., 2019; Philip et al., 2018; Uhe et al.,
2018) lasting 18-24 months was immediately followed by widespread floods (Kilavi et al., 2018; Njogu, 2021). During this time both countries also grappled with an infestation of armyworm (De Groote et al., 2020; Kumela et al., 2019) which was responsible for a reduction of food crop production. In addition to the climatic shocks and biological hazards, Kenya faced prolonged government elections that led to increased government expenses, violence and unrest. In Ethiopia, the situation was exacerbated by civil unrest and ethnic violence. These compounding factors heightened the vulnerability of communities in

both countries culminating in a humanitarian crisis, with four million people under food insecurity in Kenya (FEWS NET, 2018) and eight million people in Ethiopia (FEWS NET, 2019).

Changing climate is increasingly recognised as a health crisis (Stalhandkse et al., 2025) as it is expected to exacerbate 58% of human infectious diseases, with vector and waterborne diseases being the most affected (Mora et al., 2022). In addition to the
climatic conditions, the probability of a disease outbreak following a hazard is influenced by underlying dynamics of socioeconomic vulnerability (Jutla et al., 2013, McMichael 2009, Aitsi-Selmi and Murray 2016, Mazdiyasni and AghaKouchak 2020). Socially vulnerable populations are being disproportionately affected by the mortality associated with climate change impacts (Agache et al. 2022).

The COVID-19 outbreak increased general awareness of the challenges that arise when disasters from natural hazards and diseases collide (Tripathy et al. 2021). However, we still lack a proper understanding of the role of health, well-being, and disease outbreaks in disaster risk assessments and management. Considering the reality of rapidly changing risk dynamics (Kreibich et al. 2022), a systemic understanding of the Disaster–Disease Outbreak dynamics – i.e., the pathways through which cascading effects of extreme weather events trigger disease outbreaks and impact human health is necessary to prevent the
outbreak of diseases in the aftermath of natural hazards; develop socially-optimal and sustainable climate adaptation strategies, early warning systems, as well as relief and recovery. The examples of Mozambique, Kenya and Ethiopia demonstrate some of the health impacts of disasters and effects of consecutive disaster-disease outbreaks. These impacts will not be captured when taking a hazard-silo approach to disaster risk.

The United Nations Office for Disaster Risk Reduction (UNDRR, 2022) underscored the urgency to understand (1) changing socioeconomic vulnerability due to an earlier disaster, (2) probabilities of hazard interactions, (3) how the time-window of consecutive disasters affects impacts, and (4) the linkages between disasters, health impacts, and disease outbreaks. In response, in past years, we have seen a rise in multi-(hazard) risk studies trying to understand some of these complexities conceptually (e.g., Ward et al., 2022, Murray 2020, UNDRR 2022) and statistically (e.g., Zscheischler et al., 2018, Bevacqua
et al., 2022, De Luca et al. 2017). Moreover, De Ruiter and Van Loon (2022) discuss the great potential that exists to capture dynamics of vulnerability using existing methods used in neighbouring research fields such as compound events and socio-hydrology to capture other risk dynamics. Recently, the disaster risk field has made substantial steps forward to develop increasingly comprehensive risk assessments, accounting for the incidence of multiple hazards, trickle-down effects of cascading disasters and/or impacts, and spatiotemporal dynamics (e.g., Sett et al. 2024, Jato-Espino et al., 2025, Xoplaki et al.,
2025). A major challenge in modelling the co-occurrence of disasters lies in the misalignment of spatial and temporal scales between different hazard types and their associated impacts. Hazards such as earthquakes, floods, wildfires, or storms may occur concurrently or sequentially, but with varying onset times, durations, and spatial footprints (Gill and Malamud 2014). This makes it difficult to capture their combined consequences using standard modelling approaches that are often optimized for single hazards. Data availability and model resolution frequently constrain our ability to detect and represent compound or
cascading impacts, particularly when interactions occur across administrative boundaries or involve delayed, indirect consequences (Hillier et al., 2020). Moreover, even when hazards occur in close succession or proximity, their impacts may interact in nonlinear ways (Ridder et al. 2022).

In addition to the multi-hazard dynamics, the importance of accounting for the temporal dynamics of socioeconomic
vulnerability has been underscored in recent literature (e.g., De Angeli et al., 2022, Mora et al., 2022, Matanó et al., 2022, Kelman, 2020, Drakes and Tate 2022). Nonetheless, while in recent years many studies have focused on compound hazards (Ridder et al., 2020, Cutter 2018, Leonard et al., 2014, Zscheischler et al., 2020), the dynamics of vulnerability remain the least understood component of risk (Simpson et al., 2021, Drakes and Tate 2022, Hagenlocher et al., 2019). Owing to the complexity of health impacts, they result in heterogeneous outcomes at individual levels, requiring adaptation measures to be
precisely based on time, place and context. Hence, understanding and modelling vulnerability dynamics is a critical component to develop a systemic understanding of the Disaster–Disease Outbreak dynamics and their consequences on human health. The importance of accounting for health-related outcomes is acknowledged by the Sendai Framework for Disaster Risk Reduction (SFDRR; 2018) but it typically remains unaccounted for in risk assessments (Mazdiyasni and AghaKouchak 2020, Tilloy et al., 2019).

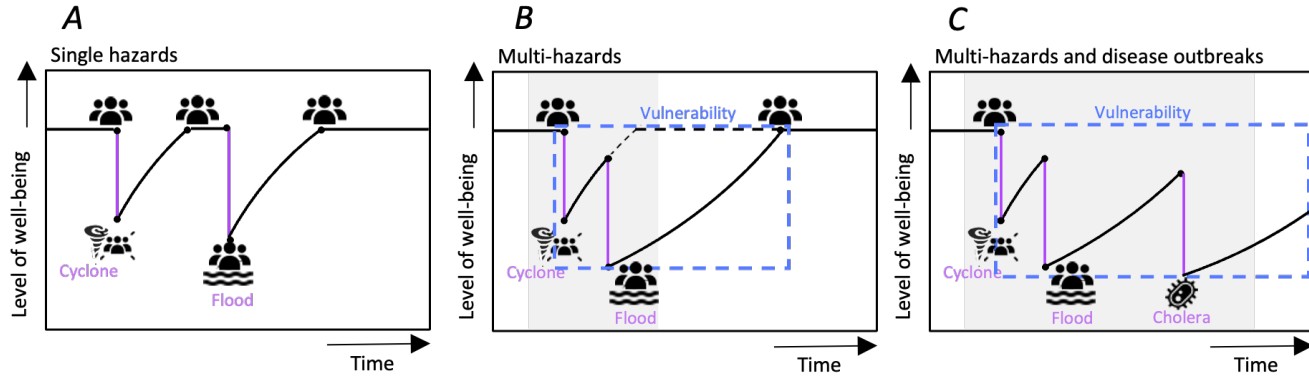


**Figure 1: Increasing disaster-risk complexity.** The figure shows from left to right increasing complexity from a typical single hazard perspective of impacts of hazards on the level of well-being (Panel A), to a multi-hazard perspective (Panel B), to the inclusion of disease outbreaks (Panel C).

In this perspective paper, we propose an initial research agenda that (1) collaborates with compound risks and socio-hydrology community to advance the modelling of occurrence probabilities and temporal element (lag times) of disasters and health/disease-outbreaks, (2) develops quantitative health risk metrics to be integrated within conventional risk management frameworks; (3) Identifies potential data sources and develops approaches to identify and map the role of stressors such as local socio-economic contexts (e.g., political instability, limited access to WASH infrastructure), interventions (e.g., Nature Based Solutions such as blue and green roofs that create vector breeding grounds (Krol et al., 2024) and their effects on the health of the affected populations. The research agenda is imperative to not only advance our systemic understanding of the Disaster–Disease Outbreak dynamics but also, enhance our modelling capabilities of the complexities of disaster risk and support the much-needed integration of public-health emergencies into risk assessments, as called for in recent scientific literature (Hillier et al., 2020, AghaKouchak et al., 2020, Simpson et al., 2021) and in recent international agreements and reports (UN 2015, WHO 2019).

## 2 State of the art and challenges

### 2.1 Co-occurrence of disasters and diseases

Systematic review of evidence from the public health impacts of disasters such as earthquakes, floods and tropical cyclones underscores the heterogeneity of disaster impacts, which are shaped not only by hazard intensity but also by variations in shelter conditions, access to healthcare, and the broader socio-environmental context (Mavrouli et al. 2023, Waddell et al. 2021). The cascading impacts such as the likelihood and intensity of disease emergence depend on a complex interplay of pre-existing health vulnerabilities, the primary and secondary effects of the disaster, and the physiographic and socio-economic characteristics of the affected area (Mavrouli et al. 2023). Despite the high prevalence of disease outbreaks post disasters, predictive modelling in this regard remains limited, with most insights derived from retrospective case reports rather than anticipatory frameworks (Alcanya et al. 2024). Consequently, preparedness and response efforts often rely on the presence and capacity of disease surveillance systems, which may be fragmented or absent in disaster-prone regions. In light of these limitations regarding predictive health-risk modelling, a promising avenue for advancing disaster risk frameworks lies in adapting methods from the growing field of compound hazard research. To understand the complex temporal dynamics between disasters and disease outbreaks, and to account for local socioeconomic circumstances that contribute to a community's vulnerability (Fig.1), we require methods to assess their dependency and interactions over time.

In recent years, compound hazard research has advanced multivariate-statistical methods, including Bayesian Networks (BNs) (e.g., Sperotto et al., 2017), to quantify hazard dependencies and joint probabilities of co-occurring disasters (Raymond et al., 2020; Tilloy et al, 2019; Hagenlocher et al., 2019; Drakes and Tate 2022; Ridder et al., 2020). These studies focus on a single hazard and co-occurring (climate-) drivers (e.g., Couasnon et al., 2020; Paprotny et al., 2020; Wahl et al., 2015; Mazdiyasni and AghaKouchak 2020; Moftakhari et al., 2019) or joint hazards such as droughts, heatwaves, and fires (e.g., Raymond et al., 2020; Matthews et al., 2019; Sutanto et al., 2020; Zscheischler and Sereviratne 2017).

These methods have also been recommended to predict disease outbreaks after a natural hazard (e.g., Tilloy et al, 2019; Hashizume et al., 2008). Despite their limited application in multi-risk, recent studies demonstrated the promising use of BNs to: (1) capture the complexities and dependencies of multi-risk due to their ability to include numerous variables with multiple dependencies, and (2) model the probability of impact chains caused by interactions between multiple variables (Sperotto et al., 2017; Tilloy et al., 2019; Liu et al., 2015; Marzocchi et al., 2012). A key limitation in using BNs for multi-hazard risk has been the challenge to incorporate temporal dynamics and feedback loops (Sperotto et al., 2017 and Tilloy et al 2019). However, Khakzad (2015) demonstrated for a risk analysis of chemical plants that this limitation can be overcome by developing a Dynamic BN (DBN). A DBN relates variables to each other over sequential time steps which enables the modelling of time dependencies and complex interactions between variables while accounting for cascading effects (Khakzad, 2015). Additionally, causal models such as SEM - Structural Equation Modelling, though data-intensive contribute to identifying the drivers and pathways including the mediating effects (Lin et al. 2017).

In recent years, machine learning (ML) and artificial intelligence (AI) techniques have emerged as powerful tools for modelling hazard co-occurrence as it allows processing increasingly large and heterogeneous datasets (Ferrario et al. 2025). By leveraging historical data, these methods can uncover complex non-linear, spatial and temporal patterns of multi-hazard events and reveal correlations across spatial and temporal scales (Pugliese Viloria et al., 2024; Reichstein et al., 2019). ML and AI-approaches have also started to be applied in for example, the modelling of infectious disease epidemics (Bauskar et al. 2022; Kraemer et al., 2025).

However, several key challenges remain. First, multivariate- statistical methods need long-term, high-resolution, and spatiotemporally explicit data (Tilloy et al., 2019); and ML/AI methods pose an additional requirement of large volumes of well-annotated training data, which may not be available for rare hazard combinations or in data-sparse regions (Ferrario et al. 2025). Next, while several studies are looking into increasing the interpretability of the predictions and underlying physical processes, this remains an ongoing challenge (e.g., Castangia et al., 2023). Nonetheless, recent studies do show promising data availability and methodological advances. Claassen et al. (2023) developed a global database of individual hazards and their consecutive occurrence. In recent years, global datasets on vector and waterborne diseases, WASH indicators, and socioeconomic indicators have increasingly become available, such as the Surveillance Atlas of Waterborne and Infectious Diseases (European Center for Disease Prevention and Control 2023), Burden of Waterborne Disease Estimates (Centres for Disease Control and Prevention 2023), WHO's WASH-database (WHO/UNICEF Joint Monitoring Programme 2024) and UNDP's HDI-database (UNDP 2023). Combining innovative modelling methods from natural hazard risk research with these available datasets will potentially contribute to extracting meaningful insights into the co-occurrence of disasters and diseases.

## 2.2 Health impacts in risk management frameworks

State-of-the-art risk assessment frameworks for natural hazards integrate hazard, exposure, and vulnerability components to estimate risk, often expressed in economic terms such as Expected Annual Damage (EAD) and Value at Risk (VaR) (Sairam et al., 2019, Steinhausen et al. 2021, Ye et al. 2024). However, adaptation decisions based solely on these metrics often fail to account for non-economic dimensions, including environmental, social, and health impacts. In some cases, the number of exposed individuals - a simplistic measure of human exposure is reported alongside economic losses (Alfieri et al., 2015; Scheiber et al. 2024).

The majority of the studies addressing negative health outcomes due to natural hazards either review past reports on impacts such as fatalities, injuries and spread of diseases (Stanke et al 2012, Kouadio et al. 2012, Suk et al. 2020, Charnley et al. 2021)

or conduct empirical analysis correlating disease trends to climate or hazard variables (Lo Iacono et al. 2017, Wu et al. 2016, Foudi et al. 2017). A very few longitudinal studies control for confounding factors (Walker-Springett et al. 2017, Bubeck et al. 2020) and quantify the effectiveness of post-disaster relief and response (Apel & Coenen, 2020). Indicators of prevalence of diseases such as risk ratio, odds ratio and incidence rate (Lee et al. 2020, Paranjothy et al. 2011) are commonly regressed against climate variables such as temperature, precipitation and socio-economic indicators such as income and gender (Speis

et al. 2019, Paranjothy et al. 2011). Although significant correlation may be revealed among these attributes, they do not contribute to process/causal understanding of the pathways through which cascading effects of disease outbreaks triggered by hazards and the impact on human health.

State-of-the-art disaster risk assessments have been increasingly incorporating semi-quantitative indicators to evaluate health

impacts—for instance, the number of affected health centres (Abbas & Routray, 2013). National risk assessments, such as those conducted by the Norwegian Directorate for Civil Protection (DSB), also apply semi-quantitative metrics, including counts of injuries and illness categories, based on subjective definitions (DSB, 2014). These existing approaches can be further enriched by incorporating more standardized and quantitative metrics such as health care expenses and metrics such as Disability Adjusted Life Years (DALY), which combine Years of Life Lost (YLL) due to fatalities and Years Lived with

Disability (YLD) (Chatterton et al. 2010, Huynh et al, 2024). Additionally, Indicators such as health-related quality of life (HRQoL), perceived recovery, and wellbeing are increasingly used to quantify the broader public health impact of disasters (Liang et al. 2014). However, these outcomes may be further modulated by socio-economic disparities (Kino et al. 2023, Sairam et al. 2025). Though these metrics enable more robust and comparable assessments, they remain largely absent from practice-oriented risk assessment frameworks. Disaster impacts, both socio-economic and health related, often create cascading

effects that worsen the overall consequences (Charnley et al. 2021), for example, the cascading effects of the immediate and short-term impacts such as injuries or financial losses on long-term physical health impacts such as musculoskeletal and cardiovascular diseases and psycho-social impacts such as depression and Post-Traumatic Stress Disorder (PTSD) (Berry et al. 2018). Since health impacts are commonly only reported at the regional or national scale (Lee et al. 2020), it is challenging to attribute these impacts to mentioned drivers that are heterogeneous at the micro-scale (Beltrame et al. 2018). Hence, the

drivers and processes leading to health risk dynamics are not widely analysed systematically alongside climate processes (Berry et al. 2018) and state-of-the-art impact metrics can rarely capture these complex cascading impacts. Ongoing efforts to integrate health metrics in disaster risk assessment frameworks include the Disaster Resilience Scorecard for Cities: Public Health System Resilience – Addendum which integrates health system resilience into urban disaster planning. It provides a structured framework with 23 indicators to evaluate the capacity of health systems to prepare for, respond to, and recover from

disasters. The tool emphasizes multi-hazard scenarios—including epidemics, infrastructure failures, and indirect health impacts—while considering vulnerable populations and continuity of care. It also assesses coordination across sectors and the ability to adapt and learn post-disaster, ensuring health systems are not isolated but integral to overall disaster resilience strategies. Though health metrics and standardized tools for disaster risk assessment are emerging, a significant gap remains in their widespread adoption, and a comprehensive, systematic framework is still needed to fully capture the intricate and

cascading impacts of natural hazards on human health. Filling this gap necessitates a shift towards multi-hazard risk management, which can account for the interconnected challenges of disasters and health crises. This approach requires understanding and managing risk across multiple threats, including both natural hazards and disease outbreaks, to ensure more robust and holistic interventions.

## 3 Research agenda and knowledge transfer

Impacts of recent disasters have demonstrated the clear need to better understand and model the interactions between disasters, disease outbreaks, and to account for health impacts of disasters. Therefore, we recommend the following research agenda, which is relevant for scientists seeking to enhance risk modelling capabilities, as well as for decision-makers and practitioners tasked with anticipating and addressing the growing complexity of disaster risks.

Modelling the probability of co-occurrence of disasters and disease outbreaks is critical for forecasting the impacts of disasters compounded with health crises. Such modelling is imperative to prevent and prepare for the outbreak of diseases following

disasters. A potential direction is to adopt the methodological advances from neighbouring fields such as multi-hazard modelling that capture interactions and feedback across disasters by utilizing the increasingly available large-scale databases. Methodological approaches include copulas, (dynamic) Bayesian networks, event coincidence analysis, and other multivariate statistical analysis (Tilloy et al. 2019).

In addition to the probability of co-occurrences of disasters and diseases, the socio-economic dimension of the affected populations plays a critical role in making them susceptible to disasters and diseases. Hence, we need comprehensive mapping of the socio-economic attributes of the populations along with post-disaster relief and recovery pathways considering scenarios of successive disaster and disease occurrences (Kouadio et al. 2012, Suk et al. 2020). The consideration of successive disaster–disease scenarios requires adopting several methods innovated by the socio-hydrology community – for instance, mechanistic models with storylines that are supported by empirical evidence and information obtained through expert knowledge (in the form of informative priors in Bayesian models – Barendrecht et al. 2019). Mechanistic models help identify pathways consisting of drivers and feedback of cascading impacts in the disaster-human-health system (Beltrame et al. 2018). They also facilitate the simulation of counterfactual scenarios which help conceptualize different intervention strategies (Adshead et al 2019). In order to conceptualize adaptation strategies at local- and region-levels, interventions from both public health (e.g., health-behaviour, education and training, supportive counselling) and disaster risk management (e.g., risk transfer, institutional framework, disaster risk reduction policy) needs to be identified and evaluated. A systemic understanding of the disaster-human-health system in the context of financial and social capacity to cope, comorbidity and existing institutional framework is pertinent to develop socially-optimal interventions (Savigny & Taghreed, 2009).

In addition to improving process understanding, exploring the use of different data types and sources would support disaster risk reduction in data-sparse situations. Health impacts of disasters are typically assessed using reported information and survey data. However, these data collection methods are both time-consuming and resource-intensive. Since disaster-specific impact data is highly personal and sensitive, researchers must comply with data privacy regulations, seek approval from ethics committees, and carefully plan fieldwork to avoid disrupting recovery efforts. Surveys offer only a time-specific snapshot of society, failing to provide continuous monitoring of the evolving situation. Given these challenges and limitations, it is crucial to explore alternative data sources to better understand the relationship between disasters, diseases, and human health systems. For example, data sources such as remote sensing and Earth Observation (EO) data show promising results to assess environmental health hazards as it can for example be used to detect damages WASH infrastructure or to identify long-standing flooded areas which in turn have a higher risk of waterborne disease outbreaks or can turn into mosquito breeding sites. Sogno et al. (2022) used EO data along with other publicly available datasets to map environments that impact public health, in specific the risk of myocardial infarction. As these data types tend to cover large temporal and spatial scales, they are explicitly useful for the assessment of the interactions between environmental factors and disaster impacts (Van Maanen et al. 2024). For example, Nusrat et al. (2022) used EO data to forecast the risk of waterborne diseases after disasters and Shah et al. (2023) conducted a literature review on the use of EO data for the mapping of WASH-related infrastructure and quality.

Leveraging these diverse data sources (e.g., Surveys, EO) and developing such mixed-method (model and data-driven) approaches requires transdisciplinary knowledge from Natural Sciences, Public Health and Social Sciences that can be used by these different sub-fields without making disciplinary compromises. Rather than trying to synergise different methods, scientists need to explore opportunities to create complementary methods and approaches to better understand the interactions between disasters and diseases, and the health impacts of disasters. In this respect, we have conceptualized a research agenda (Table 1). The research agenda outlined in this paper directly contributes to multi-hazard risk management by focusing on interlinked challenges for policy conceptualization and implementation. It emphasizes multi-level interventions and anticipatory action, aiming to provide a systemic understanding of how disasters, diseases, and societal vulnerabilities interact which is crucial for developing cohesive and effective strategies that prevent the maladaptation and asynergy.

**Table 1. Agenda for advancing research into the convergence of natural hazards and health crises**

| Research Question | Methods | Potential Outcomes | Example references |
|---|---|---|---|
| | | | |

| How can we model the probability of co-occurrence of disasters and disease outbreaks? | Adapted from Multi-hazard modelling - such as, copulas, (dynamic) Bayesian networks, event coincidence analysis, multivariate statistical analysis. | Improved forecasting of disaster-induced disease outbreaks, better preparedness and prevention strategies. | Sperotto et al., 2017; Tilloy et al., 2019; Liu et al., 2015; Marzocchi et al., 2012; Khakzad 2015. |
|---|---|---|---|
| What are the drivers and feedback mechanisms in disaster-human-health systems? | Comprehensive mapping of socio-economic variables, socio-hydrology approaches such as, storyline-based approaches, mechanistic models with Bayesian approaches with informative priors and empirical data. | Identification of vulnerable populations, pathways of cascading impacts, and improved intervention and post-disaster relief strategies | Kouadio et al. 2012, Suk et al. 2020; De Ruiter & Van Loon (2022); Barendrecht et al. 2019; Savigny & Taghreed, 2009. |
| What are the alternative data sources to improve disaster risk reduction in data-sparse situations? | Use of remote sensing - Earth Observation (EO) data, integration with publicly available datasets | Enhanced assessment of environmental health hazards, improved monitoring of WASH infrastructure and disease outbreak risks | Van Maanen et al., 2024; Nusrat et al., (2022); Sogno et al., (2022) |
| How can health impact metrics be integrated into disaster risk assessment frameworks? | Use of Disability Adjusted Life Years (DALY), Years of Life Lost (YLL), and Years Lived with Disability (YLD) in risk models, systematic analysis of cascading health impacts, micro-scale health risk attribution | More comprehensive assessment of disaster-related health burdens, improved policy decisions incorporating long-term health effects | Huynh et al. (2024); Chatterton et al. (2010); Scheiber et al. (2024); Sairam et al. (2025); Liang et al. (2014) |
| How can integrated frameworks for multi-hazard risk management be conceptualized and implemented to effectively address the cascading impacts of disasters and health crises on society? | Use a combination of dynamic Bayesian networks (DBNs) and event coincidence analysis; structural equation modelling (SEM); policy analysis, case studies, and expert elicitation and participatory modelling to inform DBNs and to create storylines. | The development of a conceptual framework or model that integrates health and disaster risk data, providing a holistic view of multi-hazard scenarios, and that and that support policymakers in designing interventions that prevent maladaptation and asynergy. | Schippers (2020), Sperotto et al. (2017); Tripathy et al. 2021; Krol et al., 2024; De Ruiter and Van Loon (2022); Haer and De Ruiter 2024 |

270   The research agenda, which emphasizes the need for increased understanding of disasters, diseases, and health impacts is targeted not only towards scientific advancement, but also aims to contribute to the following Sustainable Development Goals (SDGs): SDG3 (good health and wellbeing), SDG 6 (clean water and sanitation), SDG 11 (sustainable cities and communities), SDG 13 (climate action), and SDG 16 (peace, justice, and strong institutions) (see, Figure 2).

In the literature, the challenge of managing the risk of multiple hazards has been acknowledged. For example, challenges of maladaptation (Schippers 2020) and asynergy of disaster risk reduction measures when measures aimed at reducing the risk of one hazard have opposing effects on the risk of another hazard (De Ruiter et al. 2021). Recent real-world examples have demonstrated that similar challenges can arise in the case of disasters and disease outbreaks. For example, when the Philippines were hit by typhoon Goni during the Covid-19 pandemic, people were evacuated based on the typhoon track forecasts and forced to huddle together in evacuation facilities, enabling the spread of Covid (Gonzalo Ladera and Tiemroth 2021; IFRC-DREF, 2020). Our research agenda (Table 1) targeting process-based models considering societal and individual attributes accounts for the vulnerability dynamics (heterogeneity in local circumstances) within which these events take place. The role of vulnerability dynamics in developing comprehensive risk management measures and equitable adaptation is highlighted by recent research (De Ruiter and Van Loon 2022, Haer and De Ruiter 2024).

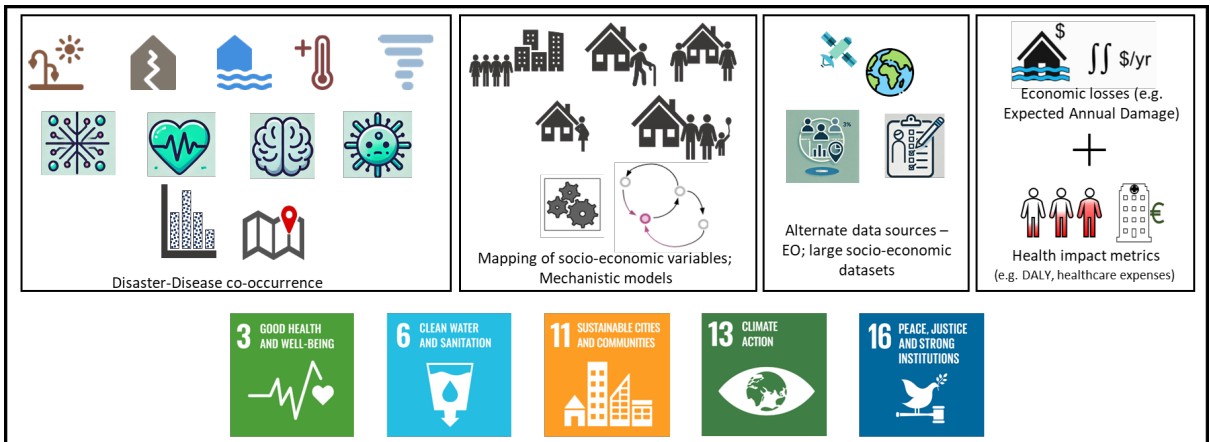

**Figure 2. Research agenda for advancing our understanding of the Disaster-Diseases and Human Health System**

## 4 Conclusions

This perspective paper underscores the urgent need to improve the integration of health impacts, disease outbreaks, into multi-hazard disaster risk assessments and management. As the frequency and complexity of concurrent disasters—such as natural hazards compounded by health crises—continue to rise, it is clear that current risk assessment models are not yet sufficiently capable to capture the full range of potential impacts. Bridging this gap requires the incorporation of novel approaches from fields such as socio-hydrology and multi-hazard modelling, which focus on understanding the interdependencies and feedbacks between disasters, diseases, and health systems.

The research agenda outlined herein highlights the importance of modelling the probability and temporal dynamics of disaster-health interactions, particularly the likelihood of disease outbreaks following natural disasters. It emphasizes the need for a more comprehensive mapping of socio-economic vulnerabilities, which influence the resilience of affected populations. By adopting mixed-method approaches that combine remote sensing data, earth observation, and empirical field data, we can enhance our ability to predict and mitigate the health impacts of disasters. The goal is not only to improve scientific understanding but also to provide actionable insights for practitioners and policymakers to create more effective and contextually appropriate interventions.

Furthermore, this agenda is aligned with key Sustainable Development Goals (SDGs) related to health, climate action, and resilience. Specifically, it contributes to SDG 6 (clean water and sanitation), SDG 11 (sustainable cities and communities), SDG 13 (climate action), and SDG 16 (peace, justice, and strong institutions), all of which require an integrated and systems-level approach to risk management.

Ultimately, this research perspective calls for a paradigm shift in disaster risk management—one that prioritizes a holistic understanding of disaster-human-health systems and leverages the full potential of interdisciplinary knowledge and technological advances. By fostering greater collaboration across disciplines and integrating health-related metrics into conventional risk frameworks, we can enhance our preparedness and response to the growing complexity of disaster risks, ensuring more resilient communities in the face of multiple, simultaneous hazards.

**Competing interests**

NS and MCdR are guest editors of special issues in NHESS.

**Acknowledgements**

MCdR received support from the MYRIAD-EU project, which received funding from the European Union's Horizon 2020 research and innovation programme under grant agreement No 101003276, and by the Netherlands Organisation for Scientific Research (NWO) (VENI; grant no. VI.Veni.222.169). NS is funded by BMFTR junior research group on Climate Environment and Health (HI-CliF) grant number - FKZ 01LN2209A.

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
