# Peer review of "Invited perspective - Redefining Disaster Risk: The Convergence of Natural Hazards and Health Crises"

_EGUsphere, 2025_

## Referee Comment (RC2)

Review of

**Invited perspective: Redefining Disaster Risk: The Convergence of**

**Natural Hazards and Health Crises**

Author(s): Nivedita Sairam and Marleen C. de Ruiter

Manuscript No.: egusphere-2025-920

Manuscript type: Invited perspectives

**General comments**

The manuscript calls for advancement in research to improve the assessment and management of natural hazards and health crises with a feasible research agenda. Looking forward to the exciting and impactful results in the future.

The manuscript proposes a research initiative on a topic within the scope of Natural Hazards and Earth System Sciences (NHESS). I would recommend this manuscript for publication with the following suggestions.

**Specific comments**

1. Widen interdisciplinary communities
   Line 15: Besides the socio-hydrology community, other interdisciplinary communities (e.g., socioeconomic, biological, climatological, geophysical, hydrological, and meteorological) have also researched the complex impacts of natural hazards and health crisis and have knowledge and experience to offer in this research agenda (e.g., compound interaction between earthquake and public health service). It is suggested to widen the lessons learned from interdisciplinary communities

2. Additional point in the research agenda towards the management side
   Abstract and Table 1: Three points and four items were listed for the call and research agenda, respectively, focusing on the multi-hazard risk assessment. I would suggest a complementary research topic focusing on "multi-hazard risk **management**", which particularly addresses the gap from science to practice, including stakeholder engagement, knowledge translation, decision science, policy analysis, applied social science research, etc. This additional point will be directly linked to Line 146-147, Line 163-165, and Line 180-185.

3. Extending the literature
   - The authors highlighted the most promising and ready technologies or tools for each point in the call. However, it is suggested to include or comment on other emergent methods and their challenges, for example, machine learning or AI for modelling of co-occurrence and data processing.

   - Ongoing efforts, particularly Disaster Resilience Scorecard for Cities: Public Health System Resilience – Addendum (https://mcr2030.undrr.org/public-health-system-

[resilience-scorecard](resilience-scorecard)), are made to improve the health emergency and disaster risk management, the subject/issues and question/assessment listed in this Public Health Scorecard are very relevant and inspiring for the section "Health impacts in risk management framework". Suggest including this reference.

- Compared to the quantitative DALY and YLD proposed, semi-quantitative metrics have already been considered in the national risk assessments. For example, population suffered from serious injuries (i.e., Serious injuries are all injuries that are not necessarily life-threatening, but require hospital treatment and/or may cause permanent damage, such as head injuries, burns and internal injuries.) and population experienced serious illness (i.e., serious illness refers to all illnesses that are not necessarily life-threatening, but require hospital treatment and/or can cause permanent damage, such as infectious diseases and mental illnesses.) in the Norwegian national risk assessment ([https://www.dsbinfo.no/DSBno/2014/Tema/FremgangsmteforutarbeidelseavNasjonaltrisikobildeNRB/](https://www.dsbinfo.no/DSBno/2014/Tema/FremgangsmteforutarbeidelseavNasjonaltrisikobildeNRB/) in Norwegian). These semi-qualitative or quantitative metrics can be further enriched with more quantitative metrics like DALY and YLD.

4. Privacy and security of health-related data
   The use of health-related data raises concerns about its privacy and security. It is worth emphasising again that the research community will respect the data privacy regulation and laws, etc and perform ethical research for the public good.

5. The challenge in the time and spatial scale for modelling the co-occurrence of disaster could be elaborated

**Minor comments**

6. Line 32: WASH acronym first appeared but the acronym was explained later in Line 123
7. Terminology: give examples of "stressors" and "interventions" for clarification for interdisciplinary communities

---

## Author Comment (AC2)

**Rebuttal**

Title: Invited perspective: Redefining Disaster Risk: The Convergence of Natural Hazards and Health Crises
Author(s): Nivedita Sairam and Marleen de Ruiter
MS No.: egusphere-2025-920
MS type: Invited perspectives
Special issue: Indirect and intangible impacts of natural hazards
* * *
**We thank all the reviewers for their valuable comments. We will implement most of their suggestions in the manuscript. Otherwise, an explanation is provided in the response letter. The responses are in italics.**
* * *
**Reviewer 1**

Very interesting work and well explained ! It was nice and easy to read, my only comment would be maybe to split the paragraphs a bit more for the final abstract. So that it would follow kind of this shape                                                                                                    :
General Background
Specific Background + Knowledge Gap
Results
Implications
So in my opinion, it would look like this with potential additions :

Recently, the disaster risk field has made substantial steps forward to develop increasingly comprehensive risk assessments, accounting for the incidence of multiple hazards, trickle-down effects of cascading disasters and/or impacts, and spatiotemporal dynamics.

While the COVID-19 outbreak increased general awareness of the challenges that arise when disasters from natural hazards and diseases collide, we still lack a comprehensive understanding of the role of disease outbreaks in disaster risk assessments and management, and that of health impacts of disasters. In specific, the occurrence probabilities and the impacts of disease outbreaks following natural hazards are not well-understood and are commonly excluded from multi-hazard risk assessments and management.

Therefore, in this perspective paper, we call for 1. learning lessons from compound risks and the socio-hydrology community for modelling the occurrence probabilities and temporal element (lag times) of

disasters and health/disease-outbreaks, 2. the inclusion of health-related risk metrics within conventional risk assessment frameworks, 3. improving data availability and modelling approaches to quantify the role of stressors and interventions on health impacts of disasters. Based on this, we develop a research agenda towards an improved understanding of the disaster risk considering potential health crises.

This is not only crucial for scientists aiming to improve risk modelling capabilities, but also for decision makers and practitioners to anticipate and respond to the increasing complexity of disaster risk.

Hope this comment will help !

*R: Thank you for your kind words and for your comment on restructuring the abstract, we really like the idea and will implement the suggestion in the revised manuscript.*
* * *
**Reviewer 2**
**General comments**
The manuscript calls for advancement in research to improve the assessment and management of natural hazards and health crises with a feasible research agenda. Looking forward to the exciting and impactful results in the future. The manuscript proposes a research initiative on a topic within the scope of Natural Hazards and Earth System Sciences (NHESS). I would recommend this manuscript for publication with the following suggestions.

*R: Thank you for the positive evaluation of our manuscript. We will implement the suggestions in the revised manuscript. We have added some information that shows the direction of the revision as responses to the comments.*

**Specific comments**
1. Widen interdisciplinary communities
Line 15: Besides the socio-hydrology community, other interdisciplinary communities (e.g., socioeconomic, biological, climatological, geophysical, hydrological, and meteorological) have also researched the complex impacts of natural hazards and health crisis and have knowledge and experience to offer in this research agenda (e.g., compound interaction between earthquake and public health service). It is suggested to widen the lessons learned from interdisciplinary communities

*R: Thank you for the very relevant comment. We have included 'socio-hydrology' explicitly as it was one of the leading examples of interactions between hydrology and social sciences. We have now extended the state-of-the art review to include research from other communities - including research on other hazards.*
*Lines (abstract): "learning lessons from interdisciplinary communities **such as** compound risks and socio-hydrology community for modelling the occurrence probabilities and temporal element (lag times) of disasters and health/disease outbreaks"*

*In the state of the art, we will include relevant lessons from the different communities -*

*"Systematic review of evidence from the public health impacts of disasters such as earthquakes, floods and tropical cyclones underscores the heterogeneity of disaster impacts, which are shaped not only by hazard intensity but also by variations in shelter conditions, access to healthcare, and the broader socio-environmental context (Mavrouli et al. 2023, Waddell et al. 2021). The cascading impacts such as the likelihood and intensity of disease emergence depend on a complex interplay of pre-existing health vulnerabilities, the primary and secondary effects of the disaster, and the physiographic and socio-economic characteristics of the affected area (Mavrouli et al. 2023). Causal models such as SEM - Structural Equation Modelling, though data-intensive contribute to identifying the drivers and pathways including the mediating effects (Lin et al. 2017).*

*Mavrouli M, Mavroulis S, Lekkas E, Tsakris A. The Impact of Earthquakes on Public Health: A Narrative Review of Infectious Diseases in the Post-Disaster Period Aiming to Disaster Risk Reduction. Microorganisms. 2023 Feb 7;11(2):419. doi: 10.3390/microorganisms11020419. PMID: 36838384; PMCID: PMC9968131.*

*Lin L, Wang Y, Liu T (2017) Perception of recovery of households affected by 2008 Wenchuan earthquake: A structural equation model. PLoS ONE 12(8): e0183631. https://doi.org/10.1371/journal.pone.0183631*

*Waddell, S. L., Jayaweera, D. T., Mirsaeidi, M., Beier, J. C., & Kumar, N. (2021). Perspectives on the Health Effects of Hurricanes: A Review and Challenges. International Journal of Environmental Research and Public Health, 18(5), 2756. https://doi.org/10.3390/ijerph18052756*

*"Despite the high prevalence of disease outbreaks post disasters, predictive modeling in this regard remains limited, with most insights derived from retrospective case reports rather than anticipatory frameworks (Alcanya et al. 2024). Consequently, preparedness and response efforts often rely on the presence and capacity of disease surveillance systems, which may be fragmented or absent in disaster-prone regions."*

*Alcayna, T., Kellerhaus, F. and Goodermote, R. Applying Anticipatory Action Ahead of Disease Outbreaks and Epidemics: A Conceptual Framework for the International Red Cross and Red Crescent Movement. Berlin: Anticipation Hub, 2024.*

*"Indicators such as health-related quality of life (HRQoL), perceived recovery, and wellbeing—are increasingly used to evaluate the broader public health impact of disasters (Liang et al. 2014). However, these outcomes may be further modulated by socio-economic disparities (Kino et al. 2023, Sairam et al. 2025)."*

*Liang Y, Lu P. Health-Related Quality of Life and the Adaptation of Residents to Harsh Post-Earthquake Conditions in China. Disaster Medicine and Public Health Preparedness. 2014;8(5):390-396. doi:10.1017/dmp.2014.94*

*Kino, S., Aida, J., Kondo, K., & Kawachi, I. (2023). Do disasters exacerbate socioeconomic inequalities in health among older people?. International Journal of Disaster Risk Reduction, 98, 104071.*

*Sairam, N., Buch, A., Zenker, M.-L., Dillenardt, L., Coenen, M., Thieken, A. H., & Jung-Sievers, C. (2025). Health-related quality of life and everyday functioning in the flood-affected population in Germany - A case study of the 2021 floods in west Germany. GeoHealth, 9, e2024GH001135. https://doi.org/10.1029/2024GH001135*

2. Additional point in the research agenda towards the management side

Abstract and Table 1: Three points and four items were listed for the call and research agenda, respectively, focusing on the multi-hazard risk assessment. I would suggest a complementary research topic focusing on "multi-hazard risk management", which particularly addresses the gap from science to practice, including stakeholder engagement, knowledge translation, decision science, policy analysis, applied social science research, etc. This additional point will be directly linked to Line 146-147, Line 163-165, and Line 180-185.

*R: Thank you very much for this comment. In the original manuscript, we discuss briefly about the transfer to practice. However, we will directly link the relevant agenda to multi-hazard risk management in the revision by including an additional research focus on multi-hazard risk management in the context of human health. We will focus on interlinked challenges to policy conceptualization and implementation in the case of multi-hazards; also focussing on multi-level interventions and anticipatory action.*

3. Extending the literature

• The authors highlighted the most promising and ready technologies or tools for each point in the call. However, it is suggested to include or comment on other emergent methods and their challenges, for example, machine learning or AI for modelling of co-occurrence and data processing.

*R: We thank the reviewer for this valuable suggestion. We agree that emerging methods such as machine learning (ML) and artificial intelligence (AI) are playing an increasingly important role in multi-hazard risk assessment, particularly for modeling co-occurrence relationships and managing large, heterogeneous datasets. In the revised manuscript, we will include a brief discussion on the potential and current limitations of ML and AI approaches in this context.*

*Adding to section 2.1: Machine Learning and Artificial Intelligence Approaches*

*In recent years, machine learning (ML) and artificial intelligence (AI) techniques have emerged as powerful tools for modelling hazard co-occurrence as it allows processing increasingly large and heterogeneous datasets (Ferrario et al. 2025). By leveraging historical data, these methods can uncover complex non-linear, spatial and temporal patterns of multi-hazard events and reveal correlations across spatial and temporal scales (Pugliese Viloria et al., 2024; Reichstein et al., 2019). ML and AI-approaches have also started to be applied in for example, the modelling of infectious disease epidemics (Bauskar et al. 2022; Kraemer et al., 2025).*

*However, several key challenges remain. First, ML/AI methods typically require large volumes of well-annotated training data, which may not be available for rare hazard combinations or in data-sparse regions (Ferrario et al. 2025). Next, while several studies are looking into increasing the interpretability of the predictions and underlying physical processes this remains an ongoing challenge (e.g., Castangia et al., 2023). Nonetheless, recent studies do show promising methodological advances.*

*Bauskar, S. R., Madhavaram, C. R., Galla, E. P., Sunkara, J. R., & Gollangi, H. K. (2022). Predicting disease outbreaks using AI and Big Data: A new frontier in healthcare analytics. European Chemical Bulletin. Green Publication. https://doi. org/10.53555/ecb. v11: i12, 17745.*

*Castangia, M., Grajales, L. M. M., Aliberti, A., Rossi, C., Macii, A., Macii, E., & Patti, E. (2023). Transformer neural networks for interpretable flood forecasting. Environmental Modelling & Software, 160, 105581.*

*Ferrario, D. M., Sanò, M., Maraschini, M., Critto, A., & Torresan, S. (2025). Harnessing Machine Learning methods for climate multi-hazard and multi-risk assessment. EGUsphere, 2025, 1-72.*

*Kraemer, M. U., Tsui, J. L. H., Chang, S. Y., Lytras, S., Khurana, M. P., Vanderslott, S., ... & Bhatt, S. (2025). Artificial intelligence for modelling infectious disease epidemics. Nature, 638(8051), 623-635.*

*Pugliese Viloria, A. D. J., Folini, A., Carrion, D., & Brovelli, M. A. (2024). Hazard susceptibility mapping with machine and deep learning: a literature review. Remote Sensing, 16(18), 3374.*

*Reichstein, M., Camps-Valls, G., Stevens, B., Jung, M., Denzler, J., Carvalhais, N., & Prabhat, F. (2019). Deep learning and process understanding for data-driven Earth system science. Nature, 566(7743), 195-204.*

• Ongoing efforts, particularly Disaster Resilience Scorecard for Cities: Public Health System Resilience – Addendum (https://mcr2030.undrr.org/public-health-system- Page **1** of **2**resilience-scorecard), are made to improve the health emergency and disaster risk management, the subject/issues and question/assessment listed in this Public Health Scorecard are very relevant and inspiring for the section "Health impacts in risk management framework". Suggest including this reference.

*R: Thank you for the valuable and relevant reference. In the context of multi-hazard risk management - considering health impacts, we will include lines on the UNDRR scorecard:*
*"Ongoing efforts to integrate health metrics in disaster risk assessment frameworks include the Disaster Resilience Scorecard for Cities: Public Health System Resilience - Addendum. The Scorecard enhances disaster risk assessments by integrating health system resilience into urban disaster planning. It provides a structured framework with 23 indicators to evaluate the capacity of health systems to prepare for, respond to, and recover from disasters. The tool emphasizes multi-hazard scenarios—including epidemics, infrastructure failures, and indirect health impacts—while considering vulnerable populations and continuity of care. It also assesses coordination across sectors and the ability to adapt and learn post-disaster, ensuring health systems are not isolated but integral to overall disaster resilience strategies."*

*UNDRR. (2021). Disaster Resilience Scorecard for Cities: Public Health System Resilience – Addendum Version 2.0. United Nations Office for Disaster Risk Reduction (UNDRR). https://mcr2030.undrr.org/sites/default/files/2021-06/UNDRR_Public%20Health%20Scorecard%20Addendum%20v2.0_English-Jan2021.pdf*

• Compared to the quantitative DALY and YLD proposed, semi-quantitative metrics have already been considered in the national risk assessments. For example, population suffered from serious injuries (i.e., Serious injuries are all injuries that are not necessarily life-threatening, but require hospital treatment and/or may cause permanent damage, such as head injuries, burns and internal injuries.) and population experienced serious illness (i.e., serious illness refers to all illnesses that are not necessarily life-threatening, but require hospital treatment and/or can cause permanent damage, such as infectious diseases and mental illnesses.) in the Norwegian national risk assessment

(https://www.dsbinfo.no/DSBno/2014/Tema/FremgangsmteforutarbeidelseavNasjonaltrisikobildeN RB/ in Norwegian). These semi-qualitative or quantitative metrics can be further enriched with more quantitative metrics like DALY and YLD.

*R: Thank you for the comment and valuable references. In the revised manuscript, we will extend the literature on metrics to include the semi-quantitative and subjective metrics and include the UNDRR scorecard within the risk management context.*
*Lines: "State-of-the-art disaster risk assessments increasingly incorporate semi-quantitative indicators to evaluate health impacts—for instance, the number of affected health centres (Abbas & Routray, 2013). National risk assessments, such as those conducted by the Norwegian Directorate for Civil Protection (DSB), also apply semi-quantitative metrics, including counts of injuries and illness categories, based on subjective definitions (DSB, 2014). These existing approaches can be further enriched by incorporating more standardized and quantitative metrics such as DALY and YLD to enable more robust and comparable assessments. "*

*Abbas, H. B., & Routray, J. K. (2013). A semi-quantitative risk assessment model of primary health care service interruption during flood: Case study of Aroma locality, Kassala State of Sudan. International Journal of Disaster Risk Reduction, 6(6), 118–128. https://doi.org/10.1016/J.IJDRR.2013.10.002*

*DSB (2014), Fremgangsmte for utarbeidelse av Nasjonalt risikobilde (NRB). https://www.dsbinfo.no/DSBno/2014/Tema/FremgangsmteforutarbeidelseavNasjonaltrisikobildeNR B (last accessed 15/07/2025)*

4. Privacy and security of health-related data
The use of health-related data raises concerns about its privacy and security. It is worth emphasising again that the research community will respect the data privacy regulation and laws, etc and perform ethical research for the public good.

*R: Thank you! The privacy and ethical considerations regarding using health-related data will be mentioned in the revised manuscript in the Research Agenda (under the context of empirical data).*
*"Since health-related data is highly personal and sensitive, researchers must respect the data privacy regulation and laws and the topics and aims of the studies must be discussed with an ethics committee."*

5. The challenge in the time and spatial scale for modelling the co-occurrence of disaster could be elaborated.

*R: We thank the reviewer for this important comment. We agree that challenges related to spatial and temporal scales are critical when modeling the co-occurrence of disasters. In the revised manuscript, we will expand our discussion to better articulate these scale-related challenges.*

*Adding to line 75 in the manuscript:*

*A major challenge in modelling the co-occurrence of disasters lies in the misalignment of spatial and temporal scales between different hazard types and their associated impacts. Hazards such as*

*earthquakes, floods, wildfires, or storms may occur concurrently or sequentially, but with varying onset times, durations, and spatial footprints (Gill and Malamud 2014). This makes it difficult to capture their combined consequences using standard modelling approaches that are often optimized for single hazards. Data availability and model resolution frequently constrain our ability to detect and represent compound or cascading impacts, particularly when interactions occur across administrative boundaries or involve delayed, indirect consequences (Hillier et al., 2020). Moreover, even when hazards occur in close succession or proximity, their impacts may interact in nonlinear ways (Ridder et al. 2022).*

*Gill, J. C., & Malamud, B. D. (2014). Reviewing and visualizing the interactions of natural hazards. Reviews of geophysics, 52(4), 680-722.*

*Hillier, J. K., Matthews, T., Wilby, R. L., & Murphy, C. (2020). Multi-hazard dependencies can increase or decrease risk. Nature Climate Change, 10(7), 595-598.*

*Ridder, N. N., Pitman, A. J., & Ukkola, A. M. (2022). High impact compound events in Australia. Weather and Climate Extremes, 36, 100457.*

**Minor comments**
6. Line 32: WASH acronym first appeared but the acronym was explained later in Line 123
7. Terminology: give examples of "stressors" and "interventions" for clarification for interdisciplinary communities
*R: The minor comments will also be implemented in the revised manuscript.*
* * *
**Reviewer 3:**

The paper proposes a research agenda towards an improved understanding of the disaster risk considering potential health crises. On this basis, the topic is aligned with the interests of the journal. Moreover, the topic is very relevant and important, and I agree that there is a gap in the space at the interface between natural hazards and health. However, I am unsure about the goal and the scientific basis of this paper.

*R: Thank you for the comment. In the revision, we will improve the clarity of the paper and the scientific basis.*

1) authors write multiple times "we call for a research agenda": do authors call or develop the agenda? What is the (scientific) process to define the agenda?

*R: Since this is a perspective (short) paper, we have not performed a systematic literature review. However, the research agenda is developed based on reviewing the state-of-the-art literature and gaps in ongoing research. In the revision, we will improve the structure of the manuscript to reflect the scientific process and we will nuance and adjust our wording of developing an initial research agenda.*

2) two points of the anticipated research agenda are: (i) develop quantitative health risk metrics and (ii) identify potential data sources and develop approaches to identify and map the role of stressors. I was expecting clear steps for these points, whereas (at the end) the "agenda" consists in Table 1 and Figure 2

*R: We fully understand the reviewer's point. Our aim is to provide research directions, while not being too prescriptive by providing exact steps.*

3) Where is the research agenda coming from? I would expect a third column where authors list sources from where they depicted the particular item for the agenda (referring to the literature review). At the moment, the paper is quite speculative and with a somewhat shallow research base (vs a systematic, objective procedure).

*R: Thank you for the comment. We will include a reference base from where the research agenda is derived.*

If the aim of the paper is to develop a research agenda, the process to arrive to the agenda should be much more articulated, sound and motivated. At the moment, the methodological approach of the paper does not satisfy the standard of the journal – in my opinion. For example, I would have expected inputs from stakeholders (interviews? surveys?). Perhaps, this expectation was set also from the title ("Invited perspective" usually is used for external elicitation, while in this paper I am not sure who "is invited"). About this, the paper is quite limited in addressing the issue of multiple stakeholders involved in the various domains underpinning natural hazards and health. The paper's idea is excellent, however the paper is not "making it" yet.

Authors are invited to review the paper and work on the scientific depth of it. Despite the different subject, examples to which they may refer are: https://doi.org/10.1111/gcb.15569; https://doi.org/10.1111/emre.12568

*R: We thank the reviewer for their comment. We have looked at other recent NHESS agenda setting perspective papers and other invited perspective papers (as our paper is not a regular research paper) and it seems that our approach follows that of several of these other agenda setting perspective papers. As mentioned in an earlier reply, we very much agree with the reviewer and will add references to support the research agenda items. However, the NHESS description of an invited perspective also states that* **"The articles should articulate the author's perspective"**. *Finally, they are also required to add "Invited perspective" to the title of their paper.*

Minor suggestions are attached as comments in the pdf.

*R: thank you for these, we will address them in the next version of our manuscript.*